# Realization of two-dimensional discrete time crystals with anisotropic Heisenberg coupling

Eric D. Switzer [1,2,3] ✉, Niall F. Robertson [4], Nathan Keenan [4,5,6], Ángel Rodríguez-Alcaraz [7,8], Andrea D'Urbano [4], Bibek Pokharel [9], Talat S. Rahman [1,3], Oles Shtanko [10] ✉, Sergiy Zhuk [4] & Nicolás Lorente [1,7] ✉

A discrete time crystal (DTC) is an out-of-equilibrium phase of matter that spontaneously breaks discrete time-translation symmetry. Previous studies have been limited to a set of models with Ising-like couplings - and mostly only in one dimension - thus precluding our understanding of the existence (or not) of DTCs in models with more realistic interactions. In this work, by combining the latest generation of IBM quantum processors with state-of-the-art tensor network methods, we demonstrate the existence of a DTC in a two-dimensional system governed by anisotropic Heisenberg interactions. We uncover a rich phase diagram encompassing spin-glass, ergodic, and time-crystalline phases, and identify the interplay of initialization, interaction anisotropy, and driving protocols in stabilizing the DTC phase. By extending the study of Floquet matter beyond simplified models, we lay the groundwork for exploring how driven systems bridge the gap between quantum coherence and emergent non-equilibrium thermodynamics.

The mechanisms by which the laws of thermodynamics arise from the underlying unitary evolution of quantum mechanics is a topic of fundamental interest in the physical sciences. For example, it is now well understood that certain out-of-equilibrium quantum systems can indeed thermalize and hence act as their own heat bath such that local observables converge to thermal values described by a Gibbs ensemble[1,2] or a generalized Gibbs ensemble[3,4]. These out-of-equilibrium phenomena are typically addressed by considering toy-models in the context of a "quantum quench", whereby non-trivial dynamics are induced by a single rapid change of the Hamiltonian of the model[5,6]. This immediately raises the question of what might be different for the thermodynamic behavior of quantum systems outside of the quench paradigm, such as driven quantum systems[7], in which energy is constantly exchanged with the environment - a situation which may more closely reflect naturally occurring thermodynamic processes. A striking example of the fundamentally different nature of quantum systems with and without a drive is provided by the emergence of discrete time crystals (DTC)[8-11]. Unlike equilibrium phases, DTCs exhibit macroscopic manifestations of coherent quantum dynamics, challenging the conventional narrative that thermodynamic behavior universally erases quantum signatures[12].

Discrete time crystals emerge in certain periodically driven many-body-localized (MBL) spin systems[13,14], where periodic driving combines with stable long-range ordering. When the drive (nearly) flips

[1]Donostia International Physics Center (DIPC), Donostia-San Sebastián, Euskadi, Spain. [2]Nanoscale Device Characterization Division, National Institute of Standards and Technology, Gaithersburg, MD, USA. [3]Department of Physics, University of Central Florida, Orlando, FL, USA. [4]IBM Quantum, IBM Research Europe - Dublin, IBM Technology Campus, Dublin, Ireland. [5]Department of Physics, Trinity College Dublin, Dublin 2, Ireland. [6]Trinity Quantum Alliance, Unit 16, Trinity Technology and Enterprise Centre, Dublin 2, Ireland. [7]Centro de Física de Materiales CFM/MPC (CSIC-UPV/EHU), Donostia-San Sebastián, Euskadi, Spain. [8]Universidad del País Vasco/Euskal Herriko Unibertsitatea UPV/EHU. Department of Polymers and Advanced Materials: Physics, Chemistry and Technology. Faculty of Chemistry, University of the Basque Country, Donostia-San Sebastián, Spain. [9]IBM Quantum, IBM T.J. Watson Research Center, New York, NY, USA. [10]IBM Quantum, IBM Research – Almaden, San Jose, CA, USA. ✉e-mail: eric.switzer@nist.gov; oles.shtanko@ibm.com; nicolas.lorente@ehu.eus

every spin, the ordering imposes a structure on the single-period unitary evolution operator, resulting in pairs of eigenstates separated by an exact $\pi$-phase rotation that remains stable against small perturbations[8–10]. Consequently, local observables exhibit oscillations with a period that is a multiple of the driving frequency, persisting indefinitely in perfectly isolated systems. This subharmonic response represents a spontaneous breaking of discrete time-translation symmetry, analogous to the breaking of continuous spatial symmetry in conventional solid-state crystals.

In this work, we explore a new setup for observing discrete time crystals. Previously, most studies focused on one-dimensional Ising-type couplings between spins due to their lower classical simulation complexity and straightforward implementation in experimental set-ups, which often rely on linear arrays[15–26]. Building on prior work[27–30], we extend these experimental investigations to a two-dimensional system on a "decorated" (heavy) hexagonal lattice and introduce spin-flip coupling. Exploring such Heisenberg-type models is motivated by their applicability to several physical systems. Heisenberg-type interactions naturally arise in a wide range of physical systems, from single-molecule magnets and metallic chains to quantum dot-based architectures[31–38]. The ability to realize such models in different experimental settings underscores the broader relevance of our findings beyond the specific implementation on superconducting qubits.

Our quantum simulations demonstrate that, despite weaker localization and theoretical arguments challenging the existence of MBL in two dimensions[39], disordered systems exhibit resilience to weak spin-flip coupling across various initial states, at least for short times. At the same time, such strong coupling drives a transition into an intermediate ergodic phase similar to that in one-dimensional systems[40,41], see Fig. 1a. Unlike the Ising model, which is largely insensitive to the choice of initial state[24,40] and governed by discrete $\mathbb{Z}_2$ symmetry, the addition of the Heisenberg terms to the Ising Hamiltonian, even as a perturbation, exhibits continuous $U(1)$ symmetry. As a result, we find initial states which exhibit strikingly different behavior to the prototypical example of a Néel initial state or a randomly chosen product state. We show that for certain values of the spin-flip coupling, oscillations originating from a Néel state decay rapidly, whereas a fully polarized state displays a remarkably stable subharmonic response reminiscent of quantum many-body scars[42,43], including cat-scar

DTCs[44,45]. This asymmetry, reflected in the phase diagrams we present in this work, underscores the fundamentally different dynamical behaviors arising from Ising versus Heisenberg interactions.

Our model, illustrated in Fig. 1c, describes a discrete-time evolution. The state of the system is expressed as $|\psi_t\rangle = U_F^t|\psi_0\rangle$, where $t$ represents integer Floquet cycles. Here, $|\psi_0\rangle$ denotes a product state of spins, and $U_F$ is the Floquet unitary operator given by

$$U_F = U_{XXZ}^{(3)} U_{XXZ}^{(2)} U_{XXZ}^{(1)} U_X, \qquad (1)$$

expressed in terms of three sub-cycle layers $U_{XXZ}^{(k)}$ and an X-gate rotation $U_X$ as described below (see Fig. 1b for schematics). The X-gate rotation corresponds to a periodic transverse kick pulse, expressed as

$$U_X = \prod_{i=1}^{N} \exp(-i\phi X_i), \qquad (2)$$

parameterized by the X-gate angle $\phi$. The cases $\phi = 0$ and $\phi = \pi/2$ correspond to specific cases of no kick and a perfect $X$-flip of all qubits, respectively.

The sub-cycle coupling layers are constructed as a composition of non-overlapping two-qubit gates, expressed as

$$U_{XXZ}^{(k)} = \prod_{(i,j)\in G_k} U_{ij}, \qquad (3)$$

where $G_k$ represents a list of qubit pairs subjected to gates at sub-layer $k$, as illustrated in Fig. 1c. The coupling gate is an implementation of a two-qubit disordered XXZ model in the form

$$U_{ij} = \exp\left[-iJ_{ij}(\epsilon X_i X_j + \epsilon Y_i Y_j + Z_i Z_j)\right], \qquad (4)$$

where $X_i$, $Y_i$, and $Z_i$ are Pauli operators for qubit $i$, $J_{ij} = 1 + \delta_{ij}$ denotes the coupling strength, $\delta_{ij} \in [-0.5, 0.5]$ represents uniformly sampled disorder, and $\epsilon$ is the spin-flip coupling strength. The disorder in the coupling induces a many-body localization (MBL) regime, following terminology established in the literature[46,47]. In the MBL regime, disorder prevents the system from absorbing energy from the periodic drive, thereby avoiding a rapid approach to the 'infinite-temperature'

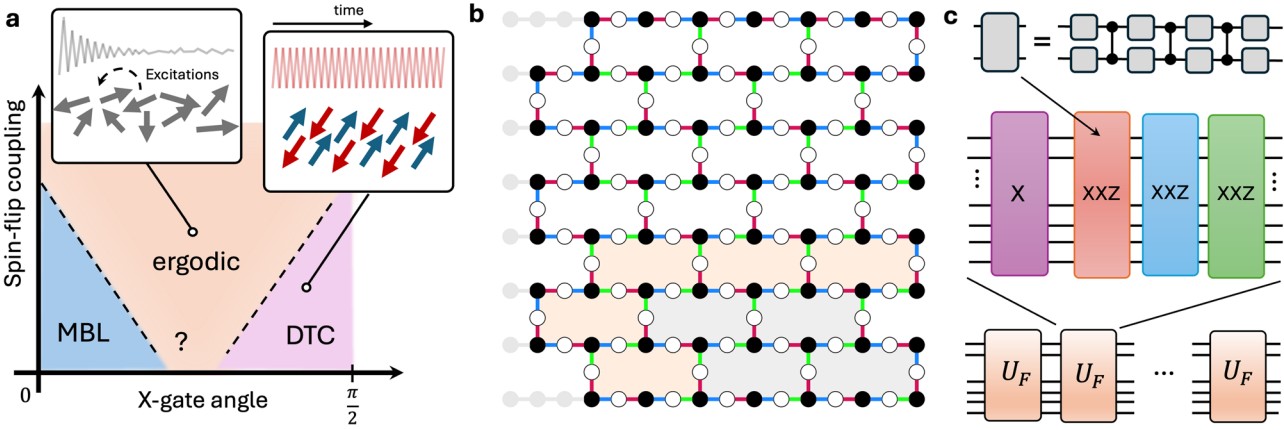

**Fig. 1 | Two-dimensional transitions driven by spin-flip coupling. a** Discrete time crystals (DTC) are stabilized through a long-range order induced by disordered Ising couplings. Introducing XY spin-flip couplings acts as an additional mechanism for state-dependent thermalization. However, the stability of time-crystal ordering in two dimensions in the presence of such coupling, along with the precise location of phase boundaries, remains an open question. **b** Circuit layout of the experimental 144-qubit system implemented on the ibm_fez device. Circles represent qubits, while lines connecting them denote two-qubit gates. Qubit colors indicate

their initial states, with the example shown corresponding to a Néel state. Edge colors encode the application order of two-qubit gates, as shown in the rightmost panel: red (1st), blue (2nd), green (3rd). Single-qubit gates are applied beforehand and their depiction is omitted for clarity. **c** Each two-qubit gate is transpiled into single-qubit gates and three controlled-Z operations, which are native to the ibm_fez device. Two-qubit gates are arranged in three layers to cover entire lattice workspace forming a Floquet cycle that repeats $t$ times.

state, characterized by trivial values of all local observables. The observation of the MBL regime can be regarded as evidence supporting the existence of a thermodynamic MBL phase, the stability of which on longer timescales in two dimensions remains debated. We use $N = 144$ qubits in two initial states: a Néel state, illustrated in Fig. 1b, and a fully polarized state.

To implement this model, we employ the 156-qubit Heron r2 quantum processor, `ibm_fez`, which uses fixed-frequency transmon qubits. From this device, we select a 144-qubit subset that includes a decorated lattice of $3 \times 7$ heavy hexagons, along with smaller configurations of $2 \times 2$ and $3 \times 3$ containing 35 and 68 qubits, respectively, as illustrated in Fig. 1b. These smaller subsystems are used to examine the scaling of order parameters with system size and to mitigate the effects of noise. Further details regarding the device architecture and implementation, including comprehensive description of the device noise, can be found in Methods and Supplementary Information, Section I.

The quantum dynamics is implemented through two sets of runs: one consisting of 50 Floquet cycles with 20,000 shots per circuit, and another comprising 30 Floquet cycles with 5,000 shots per circuit. For all experiments, the two-qubit (2Q) gate depth per cycle is 9, which is required to realize overlapping unitary operators in Eq. (4) sequentially. Consequently, the maximum circuit depth is 450 for 50-cycle experiments and 270 for 30-cycle experiments. The signal was processed using a renormalization technique that accounts for the effects of depolarization and amplitude damping. For further details, refer to the Methods section and Section III of Supplementary Information.

To verify the results of the quantum hardware, we use classical simulations based on tensor networks[48]. We consider two families of methods: Matrix Product States (MPS) and two-dimensional tensor network states. MPS-based methods are particularly well-suited to the study of one-dimensional models with local interactions. These methods also benefit from efficient contraction algorithms to calculate expectation values as well as schemes for optimal truncation of the bond dimension. However, to apply MPS-based methods to models in two dimensions, one must "unroll" the two-dimensional model into its one-dimensional representation.

To go beyond these limitations, we use two-dimensional tensor network states (2dTNS)[49] with a topology that matches our lattice, thus maintaining the locality of the model's interactions and keeping the required bond dimension low. However, unlike MPS-based methods, exact contraction of a two-dimensional network is not efficient[50] and truncations cannot be performed optimally. Instead, we use approximate methods, such as belief propagation[51]. To control the error of such approximations, we compare their results with MPS, when possible.

## Results and Discussion
### Discrete time crystalline regime
Similar to one-dimensional systems[40], we anticipate probing transitions between three distinct phases: an ordered localized spin-glass phase, a discrete time crystal phase, and a featureless ergodic phase. To differentiate these phases, we define specific observables. A defining characteristic of the discrete time crystal phase is the appearance of persistent, period-doubled oscillations in spin dynamics, accompanied by long-range spatial order. This behavior is quantified through the evolution of the time correlation function of the same qubit.

Consider an initial product state in the computational basis, i.e., $|\psi_0\rangle = |s_1\rangle \otimes \cdots \otimes |s_N\rangle$, where $s_i \in \{-1, 1\}$ specifies the spin projection along the $z$-axis. Then the time correlation function takes the form

$$\Delta(t) = \frac{1}{N}\sum_{i=1}^{N} \langle\psi_0|Z_i(0)Z_i(t)|\psi_0\rangle = \frac{1}{N}\sum_{i=1}^{N} s_i\langle\psi_t|Z_i|\psi_t\rangle, \quad (5)$$

where $N$ is the total number of qubits, the integer $t$ denotes the number of Floquet cycles and the operator $Z_i(t) = U_F^{t\dagger} Z_i U_F^t$ represents the Pauli-$Z$ operator in the Heisenberg picture. This measure effectively quantifies the individual spin memory of the initial state that persists over time.

The results of the time-correlation measurements in the time-crystalline regime (specifically at the point $\phi = 0.45\pi$, $\epsilon = 0.05$) are presented in Fig. 2a. Chosen to lie between the suspected transition to the ergodic regime and the Clifford (integrable) point, this point yields a robust DTC signal while keeping entanglement moderate and enabling reliable tensor-network simulations. The measured data exhibit distinct double-periodic oscillations, which gradually decay over time due to hardware noise, in contrast to the idealized classical simulations. The presence of double-periodic oscillations is further corroborated by the Fourier transform plot shown in the right panel of Fig. 2a, which reveals a pronounced peak at the frequency $\omega = \pi$. To mitigate the effects of noise, we employ the renormalization technique described in Eq. (9) to recover the noiseless value of $\Delta(t)$. Specifically, parameters $c(t)$ and $c'(t)$ in Eq. (9) were extracted from data obtained on a $3 \times 3$ system and subsequently applied to the full $3 \times 7$ system. The recovery of the original peak is also apparent in the frequency components. Notably, we find that a single disorder realization suffices to capture the mean of single-qubit observables, owing to the large number of qubits employed. To illustrate this, we plot the error of the mean from Eq. (5) as a shaded band in Fig. 2a, b, which remains relatively small compared to the signal.

In addition to examining time-correlation functions, we also analyze spatial correlations. We define a same-time spin correlation order parameter

$$\chi(t) = \frac{1}{M}\sum_{\langle i,j\rangle} \langle\psi_t|Z_iZ_j|\psi_t\rangle^2, \quad (6)$$

and the sum is taken over all $M$ nearest neighbors. The results are shown in Fig. 2b. Similar to the time correlations, the nearest-neighbor correlations decay over time due to hardware noise. This behavior is demonstrated by a comparison with the noiseless result obtained from classical simulations. To recover the noiseless values for the $3 \times 7$ system, we apply a renormalization technique analogous to that described in Eq. (9). This approach uses noise parameters extracted from a smaller $3 \times 3$ system (see Supplementary Information, Section III B for details). The results demonstrate that the system retains spin-spin ordering over time, which is a key feature for the emergence of the ordered regime. We also find that by expanding the set $M$ to all site pairs (which is equivalent to determining the Edwards-Anderson spin glass order parameter[24]), the noisy hardware results are similar in behavior to the results of Fig. 2b, see Supplementary Information, Section IV C for more details.

In addition to low-order correlations, it is instructive to examine a more complex measure that characterizes the circuit output. One such metric is the Hamming distance, which quantifies the minimum number of bit flips required to transform the measured bit string $\{s'_i(t) = \langle\psi_t|Z_i|\psi_t\rangle\}$ into the initial configuration $\{s_i\}$. The distribution of Hamming distances $\Phi(d, t)$ can be employed to characterize various correlations, and the variance of the distribution corresponds to the quantum Fisher information (QFI). The full distribution $\Phi(d)$ is obtained by sampling the bit strings $\{s'_i\}$ with appropriate probabilities, a process that is carried out using quantum hardware, see Supplementary Information, Section III C for more details.

The results for the Hamming distance distribution for a $2 \times 2$ system are presented in Fig. 2c, d. The measured distribution is visibly influenced by hardware noise. We observe that this effect can be effectively modeled as a process in which the bit strings undergo random and independent flips with a probability $p(t)$ that depends on the circuit cycle depth $t$. Using this approximation, we recover the

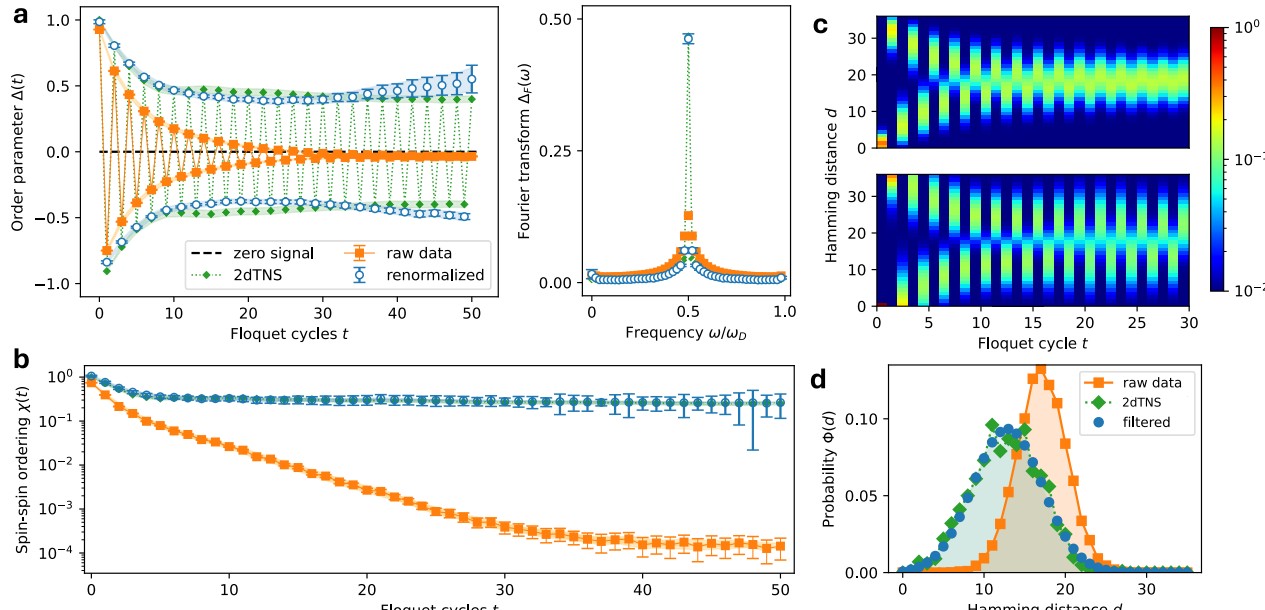

**Fig. 2 | Subharmonic behavior.** Performance of the device in the discrete time-crystalline regime. The results are compared to numerical simulations, where we use a two-dimensional tensor network state (2dTNS), see Section II B of Supplementary Information. Noise recovery is performed using the renormalization methods described in Methods and Supplementary Information, Section III. Where present, error bars indicate statistical errors corresponding to five standard deviations ($5\sigma$), estimated via a nonparametric bootstrap procedure with 200 resamples. The shaded regions represent the standard error of the mean used as the measure of finite-size error (see Methods). **a** The order parameter $\Delta(t)$, defined in Eq. (5), is plotted as a function of the number of Floquet cycles, illustrating the subharmonic response characteristic of the discrete time-crystalline regime. The data shown include raw device measurements (orange squares), classical simulations (green diamonds), and noise-recovered data (empty circles). The right panel presents the Fourier transform of the order parameter as a function of the frequency $\omega$ in units of the drive frequency $\omega_D$. The components for $\omega > \omega_D/2$ represent the original spectrum folded into this band. **b** The spin-spin correlation parameter $\chi(t)$, defined in Eq. (6), is shown. The notations are consistent with (**a**). **c** The evolution of Hamming distance distributions, defined in Eq. (S.22), is presented on a logarithmic scale for raw data (top) and mitigated data (bottom) in a $2 \times 2$ system. The mitigated distribution converges to a steady-state value. **d** An illustration of noise filtering through a comparison of Hamming distance distributions for raw device data (orange squares), 2dTNS classical simulations (green diamonds), and noise-filtered results (blue circles), exemplified at Floquet cycle 25, revealing small skewness and sizable broadening.

original distributions conditioning the recovery on their mean and variance, which correspond to the classically simulated values of the time correlation and QFI (see Supplementary Information, Section III C for more details). The experiment demonstrates that the Hamming distance rapidly converges to two distinct, non-thermal distributions at odd and even periods. The shapes of these distributions are determined by the localization properties of the system. Unlike in the one-dimensional Ising model[40], the distributions are nearly Gaussian, reflecting a higher degree of thermalization and exhibiting only a small degree of skewness.

The quantity defined in Eq. (5) enables us to construct order parameters that distinguish existing regimes of the system as we vary the parameters $\epsilon$ and $\phi$ in the model specified by Eqs. (2) and (4). Recent work[30] has explored classical simulations of this problem for the special case $\epsilon = 0$. For nonzero values of $\epsilon$, quantum correlations significantly increase the complexity of the problem (see Supplementary Information, Section III B for further details). To characterize the regimes, we define two order parameters:

$$\Delta_{\text{MBL}} = \frac{1}{NT} \sum_{t=0}^{T} |\sum_{i=1}^{N} s_i \langle Z_i(t) \rangle|, \quad (7a)$$

$$\Delta_{\text{DTC}} = \frac{1}{NT} \sum_{t=0}^{T} \sum_{i=1}^{N} (-1)^t s_i \langle Z_i(t) \rangle, \quad (7b)$$

where $N$ is the total number of qubits and $T \gg 1$ represents the maximum cycle depth. The parameter $\Delta_{\text{MBL}}$ reflects the stability of many-body localization. It vanishes in the ergodic regime, whereas both the

trivial spin-glass and discrete-time crystalline regimes exhibit nonzero values for any $T$. In contrast, the parameter $\Delta_{\text{DTC}}$ is expected to be nonzero only in the discrete-time crystalline regime. The results are presented in Fig. 3a for $T = 30$. The color plots reveal the presence of a glassy regime for small values of $\phi$ and a discrete-time crystalline regime for $\phi$ near $\pi/2$, both occurring at nonzero values of $\epsilon$. We provide a simple scaling analysis of the $\phi$-dependence of the order parameter in Fig. 3b. As the linear size of the system increases from 2 (in the $2 \times 2$ system) to 3 heavy hexagons (in the $3 \times 3$ and $3 \times 7$ systems), we observe an increase in the order parameter for certain values of $\phi$. This provides evidence of a transition between localized and ergodic regimes.

Another key parameter for characterizing potential phase transitions is the quantum Fisher information (QFI). The logarithmic growth of QFI has previously been used as evidence for the existence of a many-body localized phase[34,52]. The QFI values shown in Fig. 3c, provide evidence of very slow growth of entanglement within the glassy and discrete-time crystal regimes. In those regimes, the limiting value of QFI above the shot-noise limit ($F_Q = 1$) also indicates that the correlations may not have completely thermalized; see Section IV A of the Supplementary Information for more detail. In contrast, for points located in the center of the parameter space, we observe a faster rate of QFI growth, consistent with the behavior expected in an ergodic regime.

All the results discussed above are based on the Néel initial state, as illustrated in Fig. 1a. We expect that most input states will exhibit behavior similar to that of the Néel state. Trends in the correlator order parameter of Eq. (5) derived from experimental data, for several other random initial states, are presented in the Supplementary Information, Section IV B. A notable exceptions to this expectation are fully

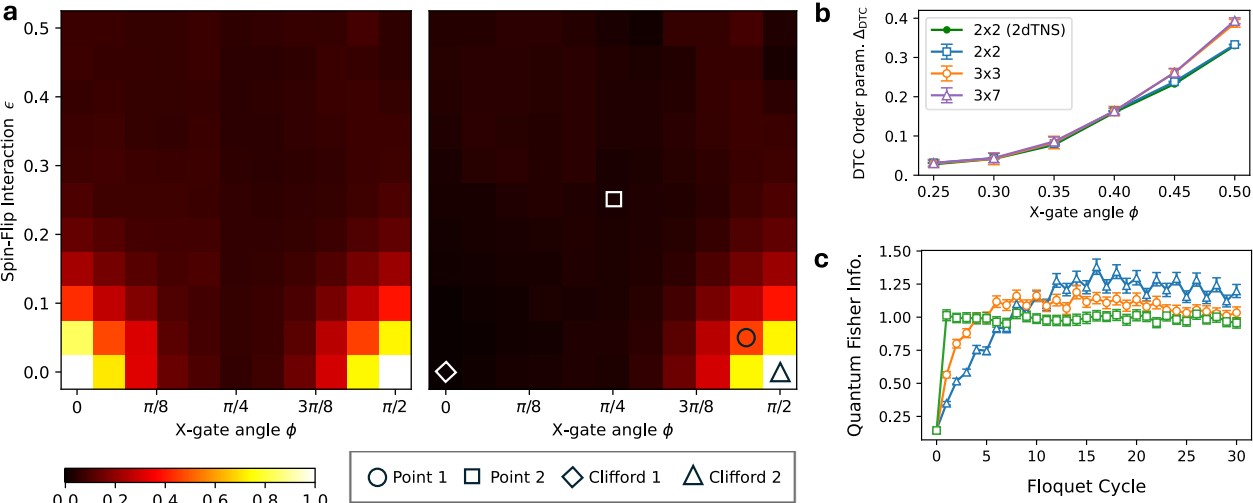

**Fig. 3 | Exploring the parameter space. a** Dependence of the MBL order parameter [Eq. (7a)] (left) and the DTC order parameter [Eq. (7b)] (right) on the spin-flip interaction strength $\epsilon$ and X-gate rotation angle $\phi$ for $T = 30$ Floquet cycles. Regions where the order parameters approach unity correspond to crossovers into the glassy localized regime (left) and the discrete time crystal (DTC) regime (right). Key reference points include: the integrable MBL Clifford point (Clifford 1), the integrable DTC Clifford point (Clifford 2), the non-integrable DTC point (Point 1), and the ergodic point (Point 2). **b** Scaling behavior of the DTC order parameter with increasing system size at $\epsilon = 0.1$, suggesting a potential crossover to the DTC regime. White dots represent normalized hardware results for $2 \times 2$, $3 \times 3$, and $3 \times 7$ systems, while filled points correspond to classical simulations for the $2 \times 2$ system. **c** Quantum Fisher information for different reference points identified in panel a, showcasing the starkly different time dependence of entanglement growth between the localized DTC regime (Clifford 2, blue triangles, and Point 1, orange circles) and the ergodic regime (Point 2, green squares). Error bars and shaded regions in (**b**, **c**) encompass the statistical error estimated by a nonparametric bootstrap procedure (see Methods). Panel (**b**) includes a $5\sigma$ error, (**c**) includes a $2\sigma$ error.

polarized states or states close to them by Hamming distance. Polarized states are distinct because they are eigenstates of the spin-flip component of the Hamiltonian in Eq. (3). Consequently, these states exhibit long-time prethermal behavior, even for $\epsilon \sim 1$. Figure 4 shows the renormalized hardware data for the polarized initial state, emphasizing the pronounced difference in the amplitude of discrete time crystal oscillations between polarized and Néel states. The observed stability of these oscillations closely resembles the behavior of quantum many-body scars[42,43] and cat-scar DTCs[44,45]. Differences in the rate of of QFI are also observed for the polarized initial state when $\epsilon$ is perturbed away from the Ising limit ($\epsilon = 0$), indicating that the symmetries of the underlying Hamiltonian play a significant role in the correlations dynamics. Further analysis derived from experimental data is presented in Supplementary Information, Section IV B.

The stability of the fully polarized state in the XXZ Hamiltonian under periodic transverse kicks raises questions about weak ergodicity breaking. The introduction of periodic kick pulses with generic phases (i.e., excluding $n\pi/2$ for integer $n$) breaks the U(1) symmetry of the XXZ Hamiltonian, potentially leading to thermalization through eigenstate mixing during Floquet evolution. Remarkably, the fully polarized state shows robustness near $\phi = 0$ and $\pi/2$, resulting in a phase diagram distinct from earlier findings[40], and this stability may stem from stable state evolution trajectories that prevent eigenstate mixing[43]. Such trajectories, resembling those in cat scar DTCs, suggest an unconventional mechanism preserving magnetic order even in the presence of perturbations[44,45].

## Outlook

So far we have considered the emergent thermodynamic behavior of a two-dimensional driven quantum system with anisotropic Heisenberg coupling. We note that there are two sources of the drive in our model: the X-gate pulse $U_X$ in Eq. (2), and the discrete time evolution of the Heisenberg model, i.e., $U_{XXZ}^{(k)}$ in Eq. (3). An interesting question therefore naturally emerges: how might the phase diagram of the kicked-Heisenberg model change if one of these drives is turned off, i.e., the

Heisenberg part of the evolution becomes a continuous time evolution operator? To probe this, one can formally define a family of models characterized by a parameter $k$ which determines the number of discrete time steps used to implement one Floquet cycle of the Heisenberg evolution. In this work, we have considered $k = 1$, while the continuous time analog would correspond to $k = \infty$. Certain regions of the phase diagram may vary as a function of $k$ - see Supplementary Information for a discussion of this point. With the quantum hardware challenges of probing larger $k$, one could instead turn to Multiproduct Formula-based methods[53–55] to study linear combinations of the results for small values of $k$, in order to gain insight to our model in Eq. (1) with continuous time evolution.

This work presents an experimental investigation of a two-dimensional discrete time crystal with anisotropic Heisenberg coupling. By developing scalable noise-renormalization techniques, we accessed the time evolution of key observables at different phases including ergodic ones, generally regarded as intractable for classical simulation methods that have rigorous error bounds, and are believed to remain challenging even for more heuristic methods where the requirement for an error bound is lifted. This allowed us to construct a detailed phase diagram of the system with regions in parameter space reflecting three regimes: one ergodic and two localized. Our findings demonstrate that the interplay between disorder and periodic driving can sustain stable and robust quantum dynamics in two dimensions. Crucially, our findings establish that many-body localization mechanisms, despite their theoretical fragility in two dimensions, can provide sufficient protection against thermalization, allowing for stable subharmonic response in a driven quantum system. Furthermore, we observed surprising scar-like behavior in the discrete time crystals, which, interestingly, is a phenomenon that appears to arise as a result of the different symmetry structure of the Heisenberg model as compared to the Ising model. This discovery provides a different perspective on weak ergodicity breaking in driven many-body systems, potentially connecting time crystals to quantum many-body scars and prethermal phases. The

methods proposed in this work open avenues for exploring large quantum systems and the emergence of exotic phases of matter on current quantum hardware even in the presence of noise.

## Methods

### Signal recovery from noisy data

The presence of hardware noise requires robust techniques for extracting signals, such as the expectation values of Pauli operators, from noisy measurements. While conventional error mitigation methods offer reasonable improvements for short-time dynamics, they fail to recover signals in circuits deep enough to explore time-crystalline behavior within our framework. To address this limitation, we introduce a physics-inspired recovery procedure that leverages specific characteristics of the noisy dynamics. In particular, we show that the influence of noise on a collective observable $O$ can be effectively modeled as

$$\langle O \rangle_{\text{noisy}} = f(t) \langle O \rangle + c(t), \tag{8}$$

where $\langle \cdot \rangle_{\text{noisy}}$ and $\langle \cdot \rangle$ denote noisy and noiseless expectation values, respectively. Here, $f(t)$ represents an amplitude attenuation factor that decays asymptotically, i.e., $f(t) \to 0$ as $t \to \infty$, while $c(t)$ denotes an offset capturing the observable's long-time behavior. We further assume that the offset exhibits a double-periodic structure, such that $c(t) = c(t+2)$. This effect can be illustrated by Fig. S3 in the Supplementary Information where the parameters of the amplitude attenuation function and the offset are estimated from the actual data.

Assuming that $f(t)$ exhibits a weak dependence on the model parameters, the observable $\langle O \rangle$ can be recovered using the following expression:

$$\langle O \rangle = \langle O \rangle' \frac{\langle O \rangle_{\text{noisy}} - c(t)}{\langle O \rangle'_{\text{noisy}} - c'(t)}. \tag{9}$$

Here, $\langle O \rangle'_{noisy}$ and $\langle O \rangle'$ represent noisy and noiseless expectation values of a "reference" observable, measured on the device for the spin-flip interaction strength $\epsilon$ and X-gate rotation angle $\phi$ corresponding to a Clifford point ($\epsilon = 0$ and $\phi = 0$ or $\phi = \pi/2$). The four free parameters, i.e., odd-period and even-period values for each offset parameter $c(t)$ and $c'(t)$, are determined via a convex optimization procedure. This procedure fits the proposed ansatz to a classical simulation of the observable on either $2 \times 2$ or $3 \times 3$ heavy hexagons. These parameters are subsequently used to reconstruct $\langle O \rangle$ on $3 \times 7$ heavy hexagons. Intuitively, this procedure cancels the multiplicative attenuation factor $f(t)$ by normalizing against Clifford point data, while the additive offsets are learned from classical simulations. For further details, as well as extensions to more sophisticated models required to recover the mean square values of two-point correlators and Hamming distances, we refer the reader to Supplementary Information, Sections III B and III C.

### Classical simulation

We employ Matrix Product State (MPS) methods by mapping the decorated hexagonal topology onto a one-dimensional chain with long-range interactions. In this approach, the unitary operators in Eq. (3) are represented as Matrix Product Operators (MPOs). We then utilize standard algorithms to contract an MPO with an MPS[48,56], enabling us to obtain a representation of the time-evolved wave function. While the bond dimension of the MPS representing the state grows rapidly with each application of $U_F$, the bond dimensions required to express the MPOs for each $U_{XXZ}^{(k)}$ remain relatively modest. These bond dimensions primarily depend on how the two-dimensional topology is mapped onto the one-dimensional chain. A key limitation of the MPS approach is that the bond dimension of the MPO can grow exponentially with the number of overlapping bonds. We outline strategies to minimize this effect in Supplementary Information,

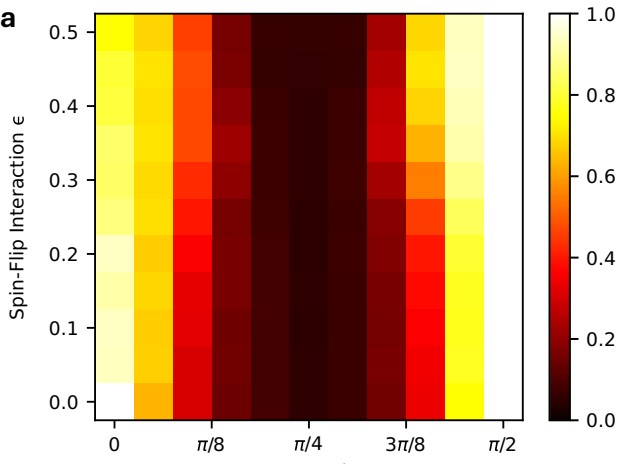

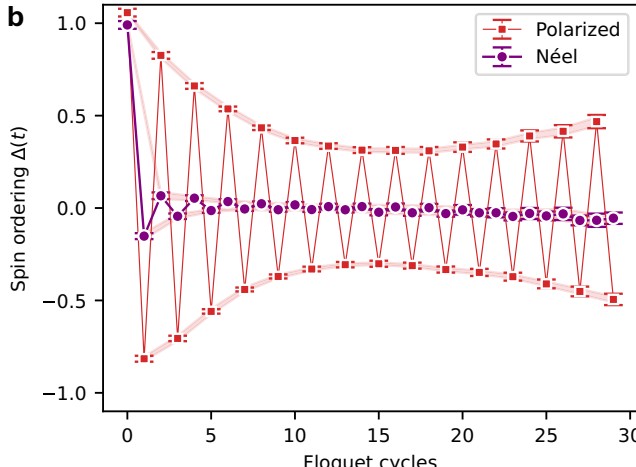

**Fig. 4 | Prethermal scar-like behavior.** Data corresponding to the polarized initial state $|\psi\rangle = |\uparrow\rangle^{\otimes N}$. **a** The MBL order parameter as a function of spin-flip interaction strength $\epsilon$ and X-gate angle $\phi$. In contrast to Fig. 3a, the order parameter remains substantially large across a wide range of $\epsilon$, indicating enhanced stability for the polarized state. **b** A comparison of the spin ordering $\Delta(t)$ as a function of Floquet cycles $t$ for the polarized and Néel initial states at $\epsilon = 0.25$ and $\phi = 0.4\pi$. The polarized state does not exhibit decay and shows robust oscillations, highlighting its prethermal behavior. Error bars in (**b**) encompass the statistical error of $5\sigma$ as estimated by a nonparametric bootstrap procedure; shaded region shows the error of the mean used as the measure of finite-size error (see Methods).

Section II A; see also ref. 57. This method is implemented using the TeNPy software package[58,59].

We also utilize classical methods that leverage a tensor network architecture matching the topology of our model[60]. Specifically, we employ an algorithm based on belief propagation[51] to determine a gauge transformation that maps the two-dimensional network into an approximate canonical form, referred to as the "Vidal-gauge" in ref. 51, inspired by related work in one dimension[61]. Single-site observables are computed by applying a rank-one approximation to the environment of the site; see ref. 51 and Section II B of Supplementary Information for further details. Higher-weight observables can also be evaluated using this approach by appending a Clifford circuit to the state and measuring the corresponding single-site operator. This procedure reproduces the expectation value of a two-site operator in the original state, albeit at the cost of increasing the entanglement within the system. For example, we employ this method to compute two-point correlation functions below. These calculations are implemented using the software package **ITensorNetworks.jl**[62], which is built on top of the package **ITensors.jl**[63].

## Implementation details

The circuits were designed and parameterized by $c(\phi, \epsilon, s, T)$ for a particular $\phi$, $\epsilon$ point on the phase diagram, where $s$ is the index of the seed used to calculate the disorder in $J$, and $T \in [0, T_{max}]$ is the number of Floquet repetitions. Thus, a total of $T_{max} + 1$ circuits are required to characterize the dynamics of each point on the phase diagram for a given disorder set. The circuits were then transpiled with the preset pass manager optimization level set to 1. This level of transpilation includes single-qubit gate optimization and inverse cancellation. The layout transpilation step was disabled to ensure a manual mapping of physical to virtual qubits. To limit the memory required for the auxiliary classical computing steps for each job after circuit submission, each set of circuits was divided into several circuit splits, for a total of $N_{cs}$ circuit splits per run. A predetermined fractional number of $N_{cs}$, along with all circuits for the two Clifford points $\epsilon = 0.0$, $\phi \in \{0, 0.5\pi\}$, were submitted using `Session` in Qiskit for a total of $N_S$ sessions. All sessions used the `SamplerV2` Qiskit primitive. For all sessions, device-level dynamical decoupling, Pauli gate twirling, and Pauli measurement twirling were disabled. Details of each run are given in Supplementary Table II of Supplementary Information.

## Determination of statistical uncertainty in experimental data

There are two sources of error relevant to assessing the significance of our results. The first is the statistical measurement error, including shot noise arising from the finite number of shots. The second stems from the variability of random-disorder realizations of the exchange-coupling strength $J_{ij}$ in Eq. (4). For all experimental data, we employ a standard bootstrap procedure applied to the data to estimate the shot noise error. Our procedure generates 200 sampled bitstring sets containing the same number of shots as the original experiment and reports this statistical error in terms of standard deviations ($\sigma$) of the resulting spread of the plotted quantity. The error is also propagated through and amplified by the renormalization procedure. To quantify the finite-size error, we report the error in the mean value for the order parameters shown in Eqs. (5) and (6), respectively, as

$$\Sigma_\Delta(t) = \sqrt{\frac{1}{N(N-1)} \sum_{i=1}^{N} (s_i \langle \psi_t | Z_i | \psi_t \rangle - \Delta(t))^2},$$

$$\Sigma_\chi(t) = \sqrt{\frac{1}{M(M-1)} \sum_{\langle i,j \rangle} (\langle \psi_t | Z_i Z_j | \psi_t \rangle^2 - \chi(t))^2}. \tag{10}$$

These quantities are expected to vanish in the limit $N, M \to \infty$, indicating self-averaging of the signal due to the large number of qubits. Values of $\Sigma_\Delta(t)$ are shown in Figs. 2a, 4b, and $\Sigma_\chi(t)$ are shown in Fig. 2b as shaded regions between $\Delta(t) \pm \Sigma_\Delta(t)$ for the polarization order parameter and between $\chi(t) \pm \Sigma_\chi(t)$ for the correlation order parameter. For visual clarity, the even and odd Floquet cycles are treated separately.

## Data availability

The data that supports the findings of this study are available from the corresponding authors on request.

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

## Acknowledgements

We thank Martin Mevissen, Sergey Bravyi, Pedro Rivero, Joana Fraxanet, Antonio Mezzacapo, and Garnett W. Bryant for helpful discussions. E.D.S., A.R., and N.L. thank the Basque Government BasQ initiative for making this project possible. E.D.S. and T.S.R. acknowledge support in part from the Department of Energy, grant number DE-FG02-07ER46354. Any mention of equipment, instruments, software, or materials does not imply recommendation or endorsement by the National Institute of Standards and Technology.

## Author contributions

E.D.S. and O.S. designed the hardware implementation, and E.D.S. performed the hardware implementation. O.S. developed the theoretical framework for analyzing the quantum dynamics presented in this work. N.F.R. and N.K. performed classical simulations, O.S. and E.D.S. performed data postprocessing, O.S. and S.Z. designed and implemented signal recovery from noisy data, A.R. and B.P. contributed to performing the experiment, and A.D. contributed to classical simulations and processing the data. N.L., O.S., S.Z., E.D.S., and T.S.R. helped guide the experiment and interpret the data. The manuscript was written by O.S., E.D.S., N.F.R., S.Z., B.P., and N.L. All authors provided suggestions for the experiment, discussed the results, and contributed to the manuscript.

## Competing interests

The authors declare no competing interests.
