## [Peer Review File · Nature Communications]

Realization of Two-dimensional Discrete Time Crystals with Anisotropic Heisenberg Coupling

Corresponding Author: Dr Eric Switzer

Version 0:

Reviewer comments:

Reviewer #1

(Remarks to the Author)

The authors use the IBM Heron r2 quantum processor to simulate discrete time crystals (DTCs) in two dimensions with a Floquet operator governed by anisotropic Heisenberg couplings. The authors observe the first signatures (the subharmonic response) of a two-dimensional DTC in the disordered case. This feature is relevant, as disorder has been hypothesized to stabilize the phase via many-body localization (MBL). The authors also map out the corresponding phase diagram as a function of spin-flip interaction and X-gate angle. These findings are made possible by extensive error mitigation and supplemented by one- and two-dimensional tensor network simulations.

This work lies at the intersection of several relevant research fields (quantum computing, dynamical phases of matter, two-dimensional many-body localization, quantum many-body scars) and demonstrates the stability of two-dimensional DTCs for the first time. In passing, the authors also note that their work shows the stability of two-dimensional MBL on the time scale of their quantum simulations. They also observe prethermal quantum many-body scars. Overall, I consider the manuscript a highly relevant piece of work.

The manuscript is well-written, accessible, and its main claims are well supported. I only have three minor comments:

1. The dots in Fig. 1b are extremely small and barely visible. I'd suggest removing them, as they don't improve the presentation.
2. In the caption of Fig. 2 and in line 128 the authors are trying to refer to a section in the Supplemental Material which is, however, missing.
3. The sentence "This behavior is depicted in Fig. 3a." in line 225 seems out of place, as the previous sentence refers to Fig. 3c.

Reviewer #2

(Remarks to the Author)

The manuscript presents an impressive experimental realization of a 2D discrete time crystal (DTC) using up to 144 qubits on IBM quantum processors. The authors investigate a 2D system governed by anisotropic Heisenberg interactions, extending previous studies that primarily focused on 1D Ising-like couplings. A key strength of this work lies in combining state-of-the-art quantum hardware with sophisticated tensor network methods and noise mitigation strategies. The comprehensive analysis reveals a rich phase diagram encompassing time-crystalline, MBL spin-glass, and ergodic phases, demonstrating the tunability of these phases through various control parameters.

It is quite impressive that the authors implement the many-body dynamics with 144 qubits up to 450 layers of two-qubit gates. We would recommend publication of this work in Nature Communications, provided that the authors satisfactorily address the following questions/comments.

Major questions:

1. All plots presented in this work, particularly those showing experimental data, do not contain error bars. The authors state that "using a single disorder realization is sufficient for describing sums of single-qubit observables, given the large number of qubits employed." We would like to see the authors verify this argument by adding error bars calculated from different qubits, or from multiple experimental runs if feasible. This is crucial for assessing the statistical significance of the observed

phenomena and the reliability of the phase boundaries.

2. The finite-size scaling in Figure 3b is inspiring. However, its generality is limited by the lack of error bars and the consideration of only two different system sizes (2×2 and 3×7). We strongly suggest the authors add data for another intermediate system size, for example, the 3×3 system, to provide more robust evidence for the finite-size scaling.

We also have some minor points that we would like the authors to respond to:

1. As the authors correctly noted, the existence of MBL in 2D systems is a controversial topic and still under debate. Therefore, we suggest modifying the sentences, for example, in line 82 "The disorder in the coupling induces a many-body localization regime," to a more cautious phrasing.
2. In Figure S14, the authors show the Edwards-Anderson spin glass order parameter for different system sizes. It appears that the order parameter decreases as the system size increases. We would expect that this order parameter would decrease to zero in the thermodynamic limit, and hence not be in the ordering phase. Is this true?
3. In Figure 3c, the QFI of the DTC phase increases to a larger value than that of the ergodic case. Could the authors provide more intuition or a physical explanation for this?
4. In a previous paper Nat. Commun. 15, 8963 (2024), the authors have implemented the 2D time-crystalline order with the planar toric code on a superconducting processor. A proper citation to this paper and some other closely related ones should be added.
5. There are some section numbers missing in the manuscript. For instance, in the Figure 2 caption, it reads, "Noise recovery is performed using the renormalization methods described in Sections ? and III of Supplementary Information." Similarly, in line 128, it states, "...be found in Supplementary Information, Sections I and ?."

Reviewer #3

(Remarks to the Author)

The manuscript reports the experimental realization of a two-dimensional discrete time crystal (DTC) with anisotropic Heisenberg coupling in a decorated hexagonal lattice system. By using 144-qubit processor experiments and two-dimensional tensor network simulation techniques, the authors successfully reveal a rich phase diagram encompassing spin-glass, ergodic, and time-crystalline phases in a two-dimensional system governed by anisotropic Heisenberg interactions with periodic driving. Experimental results demonstrate that the synergistic interplay between disorder and periodic driving can maintain stable quantum dynamical behavior in two-dimensional systems.

They introduce the spin-flip coupling to construct the anisotropic Heisenberg model, which can be obtain more interesting phenomenon compared to Ising model. They observed the fully polarized state demonstrates a stable subharmonic response that resembles the behavior of quantum many body scars including cat-scar DTCs. These achievements not only expand our understanding of nonequilibrium quantum phase transitions but also lay the foundation for investigating how driven systems establish connections between quantum coherence and emergent nonequilibrium thermodynamics.

Here I have a few questions/comments below for the authors to address.

1. Under the weak coupling condition of $\epsilon=0.05$, how can we demonstrate that the system exhibits characteristics of the anisotropic Heisenberg model rather than behavior of a perturbed Ising model?
2. The experimental results in the manuscript are primarily based on the Néel state as the initial state. For other initial states, can similar discrete time crystal phase behaviors be observed? How do different initial states influence phase diagram?
3. The manuscript repeatedly mentions the impact of hardware noise on experimental results, but lacks analysis of its specific sources and quantitative effects on the outcomes. This needs to be supplemented and improved.
4. The manuscript selects $\phi=0.45\pi$ as a representative DTC parameter (Fig 2a). Why choice this angle? Was this angle systematically optimized to maximize subharmonic response robustness?

Version 1:

Reviewer comments:

Reviewer #1

(Remarks to the Author)

The authors have satisfactorily addressed my concerns and those of the other reviewers. I therefore maintain my very positive view of this work (consistent with the other referee reports).

Reviewer #2

(Remarks to the Author)

I went through the revised manuscript, the reports from other referees and the author's replies to all three referees. In this round, the authors have made considerable efforts to improve the manuscript and address all the referees' comments and suggestions. In particular, they have added the system size 3×3 results to Fig. 3b to obtain better finite-size scaling. As far as I am concerned, they have fully addressed my concerns and their responses are satisfying. The revised manuscript has been improved substantially. I am happy to recommend its acceptance in Nature Communications with the current version.

Reviewer #3

(Remarks to the Author)

I thank the authors for their detailed and constructive responses to my previous comments, and for the corresponding revisions made to the manuscript. They have thoroughly addressed all the points raised, providing additional clarifications, data, and analysis where needed. These revisions have strengthened the manuscript and improved its clarity and rigor. This work represents a significant experimental and theoretical achievement in realizing and characterizing a two-dimensional discrete time crystal with anisotropic Heisenberg interactions on a noisy quantum processor. The successful demonstration of this non-equilibrium phase, along with the comprehensive phase diagram revealing spin-glass, ergodic, and time-crystalline regimes, provides valuable insights into the stability of quantum coherence in driven many-body systems. The observed scar-like behavior in the polarized state particularly advances our understanding of weak ergodicity breaking in two dimensions. With the issues I raised now resolved, I find the manuscript complete and interesting for publication. I am pleased to recommend its acceptance.

Response to the First Reviewer

Reviewer: *“The authors use the IBM Heron r2 quantum processor to simulate discrete time crystals (DTCs) in two dimensions with a Floquet operator governed by anisotropic Heisenberg couplings. The authors observe the first signatures (the subharmonic response) of a two-dimensional DTC in the disordered case. This feature is relevant, as disorder has been hypothesized to stabilize the phase via many-body localization (MBL). The authors also map out the corresponding phase diagram as a function of spin-flip interaction and X-gate angle. These findings are made possible by extensive error mitigation and supplemented by one- and two-dimensional tensor network simulations.*

This work lies at the intersection of several relevant research fields (quantum computing, dynamical phases of matter, two-dimensional many-body localization, quantum many-body scars) and demonstrates the stability of two-dimensional DTCs for the first time. In passing, the authors also note that their work shows the stability of two-dimensional MBL on the time scale of their quantum simulations. They also observe prethermal quantum many-body scars. Overall, I consider the manuscript a highly relevant piece of work.

The manuscript is well-written, accessible, and its main claims are well supported.”

Authors: We thank the Reviewer for the positive and constructive assessment. Below we address their comments.

Reviewer: *“I only have three minor comments:*

1. The dots in Fig. 1b are extremely small and barely visible. I’d suggest removing them, as they don’t improve the presentation.”

Authors: Following the Reviewer’s suggestion, we have removed the dots enumerating layers in Fig. 1b and adjusted the caption accordingly for legibility. The caption references the gate order as:

Edge colors encode the application order of two-qubit gates, as shown in the rightmost panel: red (1st), blue (2nd), green (3rd). Single-qubit gates are applied beforehand and their depiction is omitted for clarity.

Reviewer: *“2. In the caption of Fig. 2 and in line 128 the authors are trying to refer to a section in the Supplemental Material which is, however, missing.”*

Authors: We thank the Reviewer for catching this typo. In the revised version, we have corrected the reference in the caption of Fig. 2 as:

Noise recovery is performed using the renormalization methods described in Methods and Supplementary Information, Section III.

Reviewer: *“3. The sentence “This behavior is depicted in Fig. 3a.” in line 225 seems out of place, as the previous sentence refers to Fig. 3c.”*

Authors: We agree with the Reviewer and have removed the sentence in question because it did not properly refer to the figure.

Response to the Second Reviewer

Reviewer: *“The manuscript presents an impressive experimental realization of a 2D discrete time crystal (DTC) using up to 144 qubits on IBM quantum processors. The authors investigate a 2D system governed by anisotropic Heisenberg interactions, extending previous studies that primarily focused on 1D Ising-like couplings. A key strength of this work lies in combining state-of-the-art quantum hardware with sophisticated tensor network methods and noise mitigation strategies. The comprehensive analysis reveals a rich phase diagram encompassing time-crystalline, MBL spin-glass, and ergodic phases, demonstrating the tunability of these phases through various control parameters.*

It is quite impressive that the authors implement the many-body dynamics with 144 qubits up to 450 layers of two-qubit gates. We would recommend publication of this work in Nature Communications, provided that the authors satisfactorily address the following questions/comments.”

Authors: We are grateful to the Reviewer for their positive assessment and recommendation to publish our work. Their questions and comments are addressed below.

Reviewer: *“Major questions:*

1. All plots presented in this work, particularly those showing experimental data, do not contain error bars. The authors state that “using a single disorder realization is sufficient for describing sums of single-qubit observables, given the large number of qubits employed.” We would like to see the authors verify this argument by adding error bars calculated from different qubits, or from multiple experimental runs if feasible. This is crucial for assessing the statistical significance of the observed phenomena and the reliability of the phase boundaries.”

Authors: We thank the Reviewer for raising this point. Two independent sources of uncertainty are relevant: (i) finite-shot measurement noise and (ii) finite-size disorder variability, as pointed out by the Reviewer. Accordingly, we now include two types of error bars in Figs. 2a,b, 3b,c, and 4b. Statistical error is represented by standard error bars for all plots estimated via a nonparametric bootstrap (200 resamples) across shots for each data point. Except in Fig. 3c, these error bars show 5σ standard errors. In Fig. 3c (quantum Fisher information), we report 2σ errors due higher error values. To illustrate the finite-size error of the order parameters in Figs. 2a,b and 4b, we also plot a shaded band indicating the standard error of the mean across individual polarizations/correlators (see our changes to the Methods section quoted below). This band reflects finite-size spread in the mean of the order parameters due to the finite number of qubits and therefore quantifies the degree of disorder self-averaging.

We have modified the text accordingly as follows.

Caption of Fig. 2:

Where present, error bars indicate statistical errors corresponding to five standard deviations (5σ), estimated via a nonparametric bootstrap procedure with 200 resamples. The shaded regions represent the standard error of the mean used as the measure of finite-size error (see Methods).

Caption of Fig. 3:

Error bars and shaded regions in **b** and **c** encompass the statistical error estimated by a nonparametric bootstrap procedure (see Methods). Panel **b** includes a 5σ error, panel **c** includes a 2σ error.

Caption of Fig. 4:

Error bars in **b** encompass the statistical error of 5σ as estimated by a nonparametric bootstrap procedure; shaded region shows the error of the mean used as the measure of finite-size error (see Methods).

We have moved the modified statement about disorder self-averaging from line 116 to line 175 in the colored manuscript:

Notably, we find that a single disorder realization suffices to capture the mean of single-qubit observables, owing to the large number of qubits employed. To illustrate this, we plot the error of the mean from Eq. (5) as a shaded band in Fig. 2a,b, which remains relatively small compared to the signal.

We have added the following Methods section (see line 524 in the colored manuscript):

Determination of statistical uncertainty in experimental data

There are two sources of error relevant to assessing the significance of our results. The first is the statistical measurement error, including shot noise arising from the finite number of shots. The second stems from the variability of random-disorder realizations of the exchange-coupling strength J_{ij} in Eq. (4). For all experimental data, we employ a standard bootstrap procedure applied to the data to estimate the shot noise error. Our procedure generates 200 sampled bitstring sets containing the same number of shots as the original experiment and reports this statistical error in terms of standard deviations (σ) of the resulting spread of the plotted quantity. The error is also propagated through and amplified by the renormalization procedure. To quantify the finite-size error, we report the error in the mean value for the order parameters shown in Eqs. (5) and (6), respectively, as

$$\begin{aligned}\Sigma_{\Delta}(t) &= \sqrt{\frac{1}{N(N-1)} \sum_{i=1}^N \left(s_i \langle \psi_t | Z_i | \psi_t \rangle - \Delta(t) \right)^2}, \\ \Sigma_{\chi}(t) &= \sqrt{\frac{1}{M(M-1)} \sum_{\langle i,j \rangle} \left(\langle \psi_t | Z_i Z_j | \psi_t \rangle^2 - \chi(t) \right)^2}.\end{aligned}\tag{10}$$

These quantities are expected to vanish in the limit $N, M \rightarrow \infty$, indicating self-averaging of the signal due to the large number of qubits. Values of $\Sigma_{\Delta}(t)$ are shown in Figs. 2a and 4b, and $\Sigma_{\chi}(t)$ are shown in Fig. 2b as shaded regions between $\Delta(t) \pm \Sigma_{\Delta}(t)$ for the polarization order parameter and between $\chi(t) \pm \Sigma_{\chi}(t)$ for the correlation order parameter. For visual clarity, the even and odd Floquet cycles are treated separately.

Reviewer: “2. The finite-size scaling in Figure 3b is inspiring. However, its generality is limited by the lack of error bars and the consideration of only two different system sizes (2×2 and 3×7). We strongly suggest the authors add data for another intermediate system size, for example, the 3×3 system, to provide more robust evidence for the finite-size scaling.”

Authors: We agree that the 3×3 system size data is helpful. In the revision of our manuscript, we have added the 3×3 results to Fig. 3b. We would like to note that these data are not very different from the 3×7 data. This is likely because the relevant scaling parameter is the *linear size* of the system which is often determined by the shortest distance between opposite boundaries. Since both the 3×7 and 3×3 systems share the same linear size, the resulting curves are similar. Unfortunately, the device geometry does not allow us to measure a subset of linear size larger than three decorated hexagons.

We have added the following comment to line 225 in the colored manuscript:

We provide a simple scaling analysis of the ϕ -dependence of the order parameter in Fig. 3b. As the linear size of the system increases from 2 (in the 2×2 system) to 3 heavy hexagons (in the 3×3 and 3×7 systems), we observe an increase in the order parameter for certain values of ϕ . This provides evidence of a transition between localized and ergodic regimes.

Reviewer: “We also have some minor points that we would like the authors to respond to:

1. As the authors correctly noted, the existence of MBL in 2D systems is a controversial topic and still under debate. Therefore, we suggest modifying the sentences, for example, in line 82 “The disorder in the coupling induces a many-body localization regime,” to a more cautious phrasing.”

Authors: The Reviewer correctly notes that the existence and stability of a long-lived many-body-localized (MBL) phase in two dimensions remain under active debate. Consistent with the literature (e.g., [1, 2]), we deliberately use the term “MBL regime” as a cautious alternative to “MBL phase”. This term indicates that the observed dynamics are consistent with short-time MBL phenomenology, while asymptotic stability and a bona fide thermodynamic phase transition cannot be asserted. We agree that this clarification should be made explicit, and we have revised the text accordingly.

In the current version, we have added a footnote (see line 114 in the colored manuscript):

The existence of a thermodynamic localized phase in two dimensions is still debated. We follow the literature (e.g., [1, 2]) and use the term “regime” instead of “phase,” indicating short-time MBL phenomenology with a crossover to ergodic behavior rather than a confirmed thermodynamic transition.

Throughout the text, we have also changed the term “phase” to “regime” when describing many-body localization.

Reviewer: “2. In Figure S14, the authors show the Edwards-Anderson spin glass order parameter for different system sizes. It appears that the order parameter decreases as the system size increases. We would expect that this order parameter would decrease to zero in the thermodynamic limit, and hence not be in the ordering phase. Is this true?”

Authors: The Reviewer’s observation is correct regarding the figure displaying raw (noisy) data. The limiting value of the displayed noisy Edwards-Anderson parameter reaches zero, unlike in noiseless systems. In particular, Figure S14 shows the parameter computed for *all* spin pairs,

$$\chi_{\text{SG}} = \frac{1}{N^2} \sum_{i,j} \langle Z_i Z_j \rangle^2,$$

calculated with *raw experimental data*, reflecting noisy dynamics. In this case, the Edwards-Anderson parameter decays due to vanishing correlators $\langle Z_i Z_j \rangle_{|i \neq j}$ until it reaches a trivial $1/N$ value coming from the diagonal terms $\langle Z_i Z_i \rangle \equiv \langle I \rangle = 1$. Thus, even for large N , a zero value of the Edwards-Anderson parameter from raw experimental data is not sufficient to draw a conclusion about the actual phase of the system.

To assist readers in making the connection with the raw experimental data, we have placed the Edwards-Anderson spin glass order parameter section, including Fig. S14 and the discussion, as a subsection of a new section of the Supplementary Information, Trends in order parameters derived from raw data. We have added the above equation to this subsection and included new points in the discussion of the figure (see line 928 in the colored manuscript):

As described in the main text, if the set of spin pairs M in Eq. 6 is expanded to all qubit pairs, one recovers an Edwards-Anderson spin glass order parameter

$$\chi_{\text{SG}}(t) = \frac{1}{N(N-1)} \sum_{i,j} \langle Z_i Z_j \rangle_t^2. \quad (\text{S32})$$

The raw values of χ_{SG} are shown in Fig. S16, and are similar to the raw data shown in Fig. 2b. The parameter decays due to vanishing correlators $\langle Z_i Z_j \rangle_{|i \neq j}$ until it reaches a trivial $1/N$ value coming from the diagonal terms $\langle Z_i Z_i \rangle \equiv \langle I \rangle = 1$ in Eq. (S32). We note that because of the noisy data, a zero value of this parameter from raw experimental data (even for the large N investigated in this work) is not a sufficient condition to see the actual phase of the system.

Reviewer: “3. In Figure 3c, the QFI of the DTC phase increases to a larger value than that of the ergodic case. Could the authors provide more intuition or a physical explanation for this?”

Authors: We thank the Reviewer for this nice observation and interesting question! To address this, and in response to the Reviewer, we provide the following intuition and explanation which has been added as a new section in the Supplementary Information (see line 876 in the colored manuscript):

In our analysis of the Hamming distributions of Section III C, we define the variance of the Hamming distance distribution in Eq. (S.30), which corresponds with the quantum Fisher information order parameter $F_Q(t)$. Separating out the contributions of the diagonal elements, we obtain

$$F_Q(t) = 1 - \frac{1}{N} \sum_{i=1}^N \langle Z_i \rangle_t^2 + \frac{1}{N} \sum_{i \neq j} C_{ij}(t). \quad (\text{S31})$$

where $C_{ij}(t) = \langle Z_i Z_j \rangle_t - \langle Z_i \rangle_t \langle Z_j \rangle_t$ is two-point correlation function. The first term corresponds to $F_Q = 1$, indicating the vanishing of local observables, e.g., due to thermalization or noise. The second term is always negative and corresponds to persistent independent spin polarizations in the Z direction, while the third term is always positive and represents the spread of correlations in the system. At early times, the system is polarized and uncorrelated, which leads to $F_Q = 0$. As the system evolves, spins depolarize, making the contribution of the second term smaller, while simultaneously correlations develop, leading to $F_Q > 0$ at $t > 0$. In the ergodic phase, one can expect that both polarization and correlations vanish rapidly, resulting in $F_Q \rightarrow 1$ after a few cycles. In contrast, in a DTC regime one may expect that the correlations remain sufficiently strong and the sum of the polarization and correlation terms remains positive (assisted by the relative number of terms between the two contributions: N terms in the polarization sum and $N(N-1)$ terms in the correlator sum). The values of $F_Q > 1$ at later times shown in Fig. 3c in the main text appear to be the result of persistent system-wide correlations, characteristic of the DTC regime.

We have also added the following point to the discussion in the main text (see line 234 in the colored manuscript):

In those regimes, the limiting value of QFI above the shot-noise limit ($F_Q = 1$) also indicates that the correlations may not have completely thermalized; see Section IV A of the Supplementary Information for more detail.

Reviewer: “4. In a previous paper *Nat. Commun.* 15, 8963 (2024), the authors have implemented the 2D time-crystalline order with the planar toric code on a superconducting processor. A proper citation to this paper and some other closely related ones should be added.”

Authors: We thank the Reviewer for pointing out the work of Xiang *et al.*, which explores a distinct two-dimensional realization of time-crystalline order via a periodically driven surface code. We have added a citation to this work and a related topological-ordered DTC citation in our introduction when mentioning other two-dimensional realizations of time-crystalline order. We modified the following sentence in line 73 of the colored manuscript:

Building on prior work [3–6], we extend these experimental investigations to a two-dimensional system on a “decorated” (heavy) hexagonal lattice and introduce spin-flip coupling.

Reviewer: “5. There are some section numbers missing in the manuscript. For instance, in the Figure 2 caption, it reads, “Noise recovery is performed using the renormalization methods described in Sections ? and III of Supplementary Information.” Similarly, in line 128, it states, “...be found in Supplementary Information, Sections I and ?.”

Authors: We thank the Reviewer for spotting these typos. In the updated version, we have fixed these issues.

Response to the Third Reviewer

Reviewer: “The manuscript reports the experimental realization of a two-dimensional discrete time crystal (DTC) with anisotropic Heisenberg coupling in a decorated hexagonal lattice system. By using 144-qubit processor experiments and two-dimensional tensor network simulation techniques, the authors successfully reveal a rich phase diagram encompassing spin-glass, ergodic, and time-crystalline phases in a two-dimensional system governed by anisotropic Heisenberg interactions with periodic driving. Experimental results demonstrate that the synergistic interplay between disorder and periodic driving can maintain stable quantum dynamical behavior in two-dimensional systems. They introduce the spin-flip coupling to construct the anisotropic Heisenberg model, which can be obtain more interesting phenomenon compared to Ising model. They observed the fully polarized state demonstrates a stable subharmonic response that resembles the behavior of quantum many body scars including cat-scar DTCs. These achievements not only expand our understanding of nonequilibrium quantum phase transitions but also lay the foundation for investigating how driven systems establish connections between quantum coherence and emergent nonequilibrium thermodynamics.”

Authors: We thank the Reviewer for highlighting the significance of our experimental and theoretical work, and we respond to their questions and comments below.

Reviewer: “Here I have a few questions/comments below for the authors to address.

1. Under the weak coupling condition of $\epsilon = 0.05$, how can we demonstrate that the system exhibits characteristics of the anisotropic Heisenberg model rather than the behavior of a perturbed Ising model?

Authors: We thank the Reviewer for raising this important question. There exists measurable differences between perturbing ϕ from the integrable point $\phi = \pi/2$ (Ising perturbation) and adding a small ϵ (Heisenberg perturbation). The main distinction between these regimes lies in their symmetries. The Ising regime exhibits discrete \mathbb{Z}_2 Ising (parity) symmetry, whereas the Heisenberg regime possesses continuous $U(1)$ (charge conservation) symmetry.

The Ising regime is insensitive to the choice of initial state, a feature also noted in previous works [7, 8]. In contrast, the Heisenberg regime can strongly depend on the initial state. In particular, Fig. 4 demonstrates the difference in the order parameter between the Néel and polarized states, as well as the clear asymmetry of the phase diagram in the ϵ and ϕ directions for the polarized state (compare to Fig. 3a).

In the new version of the manuscript, we have improved the presentation of this difference, see line 85 of the colored manuscript:

Unlike the Ising model, which is largely insensitive to the choice of initial state [7, 8] and governed by discrete \mathbb{Z}_2 symmetry, the addition of the Heisenberg terms to the Ising Hamiltonian, even as a perturbation, exhibits continuous $U(1)$ symmetry. As a result, we find initial states which exhibit strikingly different behavior to the prototypical example of a Néel initial state or a randomly chosen product state. We show that for certain values of the spin-flip coupling, oscillations originating from a Néel state decay rapidly, whereas a fully polarized state displays a remarkably stable subharmonic response reminiscent of quantum many-body scars [9, 10], including cat-scar DTCs [11, 12]. This asymmetry, reflected in the phase diagrams we present in this work, underscores the fundamentally different dynamical behaviors arising from Ising versus Heisenberg interactions.

To strengthen our presentation, we have performed a diagnostic of the quantum Fisher information growth between different initial states in the Ising and perturbed-Ising (to Heisenberg, $\epsilon = 0.05$ and $\epsilon = 0.1$) models. We present these results in a new Fig. S14 in Supplementary Information (see below).

We have included a discussion to accompany this figure in the Supplementary Information (see line 895 in the colored manuscript):

For parameter ranges where the MBL and DTC regimes, identified by the two-time correlator, are shared by both the Néel (Fig. 3) and polarized (Fig. 4) initial states, noisy $F_Q(t)$ exhibits logarithmic growth. This behavior is remarkable given that these parameter ranges encompass unperturbed ($\epsilon = 0$) and perturbed ($\epsilon > 0$) kicked Ising models. The curvature of $F_Q(t)$ displays a pronounced dependence on the initial state when spin-flip components are introduced ($\epsilon > 0$), as shown in the experimental data of Fig. S14. For Floquet cycles prior to complete depolarization, the kicked Ising model in Fig. S14a shows little discernible difference between the Néel and polarized initial states as in both cases its growth originates from the underlying noise. In contrast, at values of ϵ that perturbatively move the system into the Heisenberg-perturbed regime, Fig. S14b,c, the curvature of F_Q differs dramatically between these two initial states.

Figure S14. $F_Q(t)$, given in Eq. (S.30), derived from experimental raw data for the Néel and polarized initial states, at the (ϕ, ϵ) points on the phase diagram located at **a**, (0.0, 0.0), **b**, (0.0, 0.05), and **c**, (0.0, 0.1). The shot noise limit at $F_Q = 1$ (gray line), is added for clarity.

We reference the above discussion in the main text, see line 250 in the colored manuscript:

Differences in the rate of of QFI are also observed for the polarized initial state when ϵ is perturbed away from the Ising limit ($\epsilon = 0$), indicating that the symmetries of the underlying Hamiltonian play a significant role in the correlations dynamics. Further analysis derived from experimental data is presented in Supplementary Information, Section IV B.

Reviewer: “2. The experimental results in the manuscript are primarily based on the Néel state as the initial state. For other initial states, can similar discrete time crystal phase behaviors be observed? How do different initial states influence phase diagram?”

Authors: We examine initial-state dependence in both the previous and current versions (see Fig. 4b). In particular, a fully polarized initial state (all spins aligned along $+z$) yields a phase diagram that differs markedly from that of the Néel state and displays pronounced robustness to the Heisenberg perturbation. This robustness is expected because the fully polarized product state is an eigenstate of the Heisenberg term, which suppresses its effect on the ensuing dynamics. The resulting long-lived response is reminiscent of scar-like behavior, as noted in the previous version of the manuscript.

By contrast, other initial states do not exhibit such stability and behave similarly to the Néel state. To substantiate this point, we added frequency spectra for several randomly polarized initial states to the Supplementary Information. Here “randomly polarized” denotes spins with independently sampled orientations (specified in the Supplementary Information, Section IV B). Although these supplementary plots do not use the renormalization scheme detailed in our work (and thus represent raw experimental data), the existence of the peak in the frequency plot is relatively insensitive to noise. From these data, we find that the randomly polarized initial states exhibit the same qualitative behavior as the Néel state for nonzero Heisenberg perturbation ϵ , in contrast to the fully polarized state.

We added new Fig. S15 to the Supplementary Information (see below). In addition to the figure, we have added the following discussion to Supplementary Information (see line 907 in the colored manuscript):

The presence of subharmonic responses in local observables under a periodic drive is one of the primary indicators of discrete time-crystalline behavior. A Fourier transform of these observables, such as the two-time correlator order parameter defined in Eq. (5), reveals these subharmonic responses through a pronounced peak at characteristic frequency $\omega = \omega_D/2$, where ω_D is the drive frequency. In the main text, evidence of such a subharmonic response is observed across a variety of parameter regimes and initial states, including both the Néel (Fig. 3) and polarized (Fig. 4) initial states. Notably, the polarized initial state exhibits a robust subharmonic signal over a wider range of spin-flip couplings ϵ compared to the Néel initial state. Other initial states do not display this stability and behave nearly similarly to the Néel initial state. In Fig. S15, the Fourier transforms of the two-time correlator order parameter, derived from experimental data, are shown for several random initial states alongside

Figure S15. The Fourier transform of the order parameter $\Delta(t)$, defined in Eq. (5) and derived from experimental raw data is shown as a function of the frequency ω in units of the drive frequency ω_D . The components for $\omega > \omega_D/2$ represent the original spectrum folded into this band. Different initial states are plotted, which include the Néel (blue squares) and polarized (orange circles) initial states, and three realizations of random initial states (purple inverted triangles, green diamonds, and red triangles), as described in Section IV B. Panels correspond to points on the phase diagram (ϕ, ϵ) located at **a**, $(0.45\pi, 0.05)$, **b**, $(0.35\pi, 0.15)$, and **c**, $(0.3\pi, 0.2)$. Curves for different initial states exhibit a vertical displacement for illustration purposes.

those for the Néel and polarized initial states. Each random state is generated as a product state of independently and randomly assigned maximally polarized spins $s_i \in \{-1, 1\}$. Near the Clifford point in Fig. S15a, the peak at $\omega = \omega_D/2$ is pronounced for all initial states, though the polarized initial state shows a larger amplitude than the other states. As the parameters are tuned away from the integrable point, by decreasing the kick-angle ϕ and increasing spin-flip strength ϵ , the polarized state retains a half-frequency peak, in contrast to the other initial states, see Fig. S15b,c. In Fig. S15c, as the kick angle approaches the ergodic regime at $\phi = \pi/4$ the original half-frequency peaks disappears, indicating the cessation of time-crystalline behavior.

We reference this expanded discussion in the main text, see line 240 in the colored manuscript:

We expect that most input states will exhibit behavior similar to that of the Néel state. Trends in the correlator order parameter of Eq. (5) derived from experimental data, for several other random initial states, are presented in the Supplementary Information, Section IV B. A notable exceptions to this expectation are fully polarized states or states close to them by Hamming distance.

Reviewer: “3. The manuscript repeatedly mentions the impact of hardware noise on experimental results, but lacks analysis of its specific sources and quantitative effects on the outcomes. This needs to be supplemented and improved.”

Authors: We thank the Reviewer for this suggestion. We would like to emphasize that the previous version of the manuscript does discuss the role of noise. For example, the Methods section models the effect of dephasing and amplitude damping in Eq. (8) and suggest an error mitigation scheme. We also have a Supplementary Information Section I that provides comprehensive analysis of the device noise.

We recognize, however, that we may not adequately guide the reader to the relevant parts of the paper. In the current version, we have improved this aspect. We now provide explicit references in the main text and Methods section to the Supplementary Information where the noise analysis is presented.

In addition, we have added a new figure to Supplementary Information showing the effect of noise on the actual order parameter data, derived from an analysis of Clifford dynamics. We hope that this expanded discussion provides a clear and sufficient understanding of the role of noise in our results.

We have made the following changes in response to the Reviewer’s comment. In Supplementary Information, we have added a figure illustrating the quantitative effect of noise:

We have accompanied this figure with the comments in Supplementary Information (see line 605 in the colored manuscript):

An independent, observable-dependent quantification of noise can be obtained by evaluating the observable in the Clifford regime. At this point, the noiseless values of $|\Delta(t)|$ and $\chi(t)$ in Eqs. (5) and (6) both equal unity,

Figure S3. Quantitative noise analysis at the Clifford point (marked as Clifford 2 in Fig. 3 in the main text), used as a reference for error mitigation. In noiseless simulations, the order parameter alternates as $\Delta(t) = (-1)^t$, where t is the number of Floquet cycles. With noise, the observed signal takes the form $\Delta_{\text{noisy}}(t) = \Delta(t)f(t) + \delta$, where $f(t)$ accounts for decoherence noise (dashed blue curves) and δ is an offset due to amplitude damping noise (red line); see Eq. (S.5). The solid black line corresponds to zero signal. Approximating $f(t) \approx Ae^{-t/T}$, with T the characteristic decay time, allows extraction of the parameters $A = 0.973$, $T = 14.5$, and $\delta = -0.063$ (even cycles) and $\delta = -0.019$ (odd cycles), providing a quantitative characterization of the noise. These values are subsequently used to mitigate errors at non-Clifford points (see Section III).

which allows us to isolate the impact of noise. An example of this analysis is shown in Fig. S3 for $\Delta(t)$ at the Clifford point 2 in Fig. 3 in the main text ($\epsilon = 0$, $\phi = \pi/2$). The data reveals two distinct effects of noise: (i) a decay of the signal due to coherence loss, and (ii) a systematic offset arising from the non-unital component of the noise (amplitude damping). Fig. S3 indicates a decoherence time of approximately $T \approx 15$ Floquet cycles, corresponding to $D \approx 135$ two-qubit gate layers. The offset is approximately $\delta \approx -4 \times 10^{-2}$, which requires the dedicated error-mitigation strategy described in Section III.

In Methods, we have added a reference to this figure (see line 462)

This effect can be illustrated by Fig. S3 in the Supplementary Information where the parameters of the amplitude attenuation function and the offset are estimated from the actual data.

In the main text, we have added a reference to the noise model in line 123 of the colored manuscript:

Further details regarding the device architecture and implementation, including comprehensive description of the device noise, can be found in Methods and Supplementary Information, Section I.

Reviewer: “4. The manuscript selects $\phi = 0.45\pi$ as a representative DTC parameter (Fig 2a). Why choice this angle? Was this angle systematically optimized to maximize subharmonic response robustness?”

Authors: We thank the Reviewer for raising this question. We use $\phi = 0.45\pi, \epsilon = 0.05$ as our operating point in Fig. 2 because it balances two limiting regimes. It is sufficiently detuned from the suspected phase-transition point—where the DTC order parameter is strongly suppressed—and from the Clifford (integrable) point—where the dynamics become trivial. At $\phi = 0.45\pi, \epsilon = 0.05$, the subharmonic response remains robust while entanglement growth is moderate, enabling reliable tensor-network benchmarks and a transparent comparison between experiment and classical simulation. For these reasons, we regard $\phi = 0.45\pi$ as the optimal working point for resolving DTC dynamics in our setup.

We have added the following clarification to line 164 of the colored manuscript:

Chosen to lie between the suspected transition to the ergodic regime and the Clifford (integrable) point, this point yields a robust DTC signal while keeping entanglement moderate and enabling reliable tensor-network simulations.

References

1. Morningstar, A., Colmenarez, L., Khemani, V., Luitz, D. J. & Huse, D. A. Avalanches and many-body resonances in many-body localized systems. *Phys. Rev. B* **105**, 174205 (2022).
2. Long, D. M., Crowley, P. J. D., Khemani, V. & Chandran, A. Phenomenology of the prethermal many-body localized regime. *Phys. Rev. Lett.* **131**, 106301 (2023).
3. Shinjo, K., Seki, K., Shirakawa, T., Sun, R.-Y. & Yunoki, S. Unveiling clean two-dimensional discrete time quasicrystals on a digital quantum computer (2024). ArXiv:2403.16718, 2403.16718.
4. Fernandes, L., Tindall, J. & Sels, D. Bipartite discrete time crystals on decorated lattices (2024). ArXiv:2411.00651, 2411.00651.
5. Xiang, L. *et al.* Long-lived topological time-crystalline order on a quantum processor. *Nature Communications* **15**, 8963 (2024).
6. Wahl, T. B., Han, B. & Béri, B. Topologically ordered time crystals. *Nature Communications* **15**, 9845 (2024).
7. Mi, X. *et al.* Time-crystalline eigenstate order on a quantum processor. *Nature* **601**, 531–536 (2022).
8. Ippoliti, M., Kechedzhi, K., Moessner, R., Sondhi, S. & Khemani, V. Many-Body Physics in the NISQ Era: Quantum Programming a Discrete Time Crystal. *PRX Quantum* **2**, 030346 (2021).
9. Turner, C. J., Michailidis, A. A., Abanin, D. A., Serbyn, M. & Papić, Z. Weak ergodicity breaking from quantum many-body scars. *Nat. Phys.* **14**, 745–749 (2018).
10. Maskara, N. *et al.* Discrete Time-Crystalline Order Enabled by Quantum Many-Body Scars: Entanglement Steering via Periodic Driving. *Phys. Rev. Lett.* **127**, 090602 (2021).
11. Huang, B. Analytical theory of cat scars with discrete time-crystalline dynamics in Floquet systems. *Phys. Rev. B* **108**, 104309 (2023).
12. Bao, Z. *et al.* Creating and controlling global Greenberger-Horne-Zeilinger entanglement on quantum processors. *Nat. Commun.* **15**, 8823 (2024).